# Proteomic Advances in Glial Tumors through Mass Spectrometry Approaches

**DOI:** 10.3390/medicina55080412

**Published:** 2019-07-27

**Authors:** Radu Pirlog, Sergiu Susman, Cristina Adela Iuga, Stefan Ioan Florian

**Affiliations:** 1Department of Morphological Sciences, “Iuliu Hatieganu” University of Medicine and Pharmacy, Cluj-Napoca 400012, Romania; 2Department of Pathology, IMOGEN Research Centre, Cluj-Napoca 400012, Romania; 3Department of Pharmaceutical Analysis, Faculty of Pharmacy, “Iuliu Hațieganu” University of Medicine and Pharmacy, Cluj-Napoca 400012, Romania; 4Department of Proteomics and Metabolomics, MedFuture Research Center for Advanced Medicine, “Iuliu Hațieganu” University of Medicine and Pharmacy, Cluj-Napoca 400012, Romania; 5Department of Neurosurgery, “Iuliu Hatieganu” University of Medicine and Pharmacy, Cluj-Napoca 400012, Romania

**Keywords:** diffuse glioma, glioblastoma, proteomics, mass spectrometry, biomarker

## Abstract

Being the fourth leading cause of cancer-related death, glial tumors are highly diverse tumor entities characterized by important heterogeneity regarding tumor malignancy and prognosis. However, despite the identification of important alterations in the genome of the glial tumors, there remains a gap in understanding the mechanisms involved in glioma malignancy. Previous research focused on decoding the genomic alterations in these tumors, but due to intricate cellular mechanisms, the genomic findings do not correlate with the functional proteins expressed at the cellular level. The development of mass spectrometry (MS) based proteomics allowed researchers to study proteins expressed at the cellular level or in serum that may provide new insights on the proteins involved in the proliferation, invasiveness, metastasis and resistance to therapy in glial tumors. The integration of data provided by genomic and proteomic approaches into clinical practice could allow for the identification of new predictive, diagnostic and prognostic biomarkers that will improve the clinical management of patients with glial tumors. This paper aims to provide an updated review of the recent proteomic findings, possible clinical applications, and future research perspectives in diffuse astrocytic and oligodendroglial tumors, pilocytic astrocytomas, and ependymomas.

## 1. Introduction

Diffuse astrocytic and oligodendroglial tumors are a heterogeneous group of primary brain tumors with significant differences regarding tumor malignancy and prognosis [1]. Previous research focused on decoding the genetic alterations behind these tumors, *TERT* promoter mutation, *EGFR* amplification and the presence of its active mutant *EGFRvIII*, chromosome 7 copy number alteration and *MGMT* gene promoter methylation being observed in grade IV gliomas and mutations in *IDH-1*, *IDH-2*, *ATRX*, *CIC*, *FUB*, and 1p/19q co-deletion being frequently identified in grade lower grade gliomas [2]. The identification of genetic markers stayed at the basis of the elaboration of the updated 2016 World Health Organisation (WHO) classification of tumors of the Central Nervous System (CNS), which for the first time integrated the status of the *IDH* gene and 1p/19q co-deletion into the diagnostic criteria [3].

Despite the identification of important alterations in the genome of glial tumors, a knowledge gap of the mechanisms underlying glioma malignancy still exists [4]. Genomic studies offered important insights into genetic events at the molecular level [5]. However, due to intricate mechanisms involved in glioma tumorigenesis, the identified genetic alterations do not always correlate with the functional proteins expressed in tumor cells and serum [6]. For a more comprehensive understanding of the biological processes that take place in glial tumor cells, we need to fill the gap between known genomic alterations and the proteomic pattern of gliomas in order to identify reliable predictive, diagnostic, and prognostic biomarkers [7].

Mass spectrometry (MS) is an analytical technique that measures the mass-to-charge ratio (m/z) of ions. The results are typically presented as a mass spectrum, a plot of intensity as a function of the mass-to-charge ratio. It offers the possibility of identifying multiple proteins based on their molecular mass [8]. MS-based proteomic approaches enabled researchers to identify and characterize proteomes of individual cells [9,10,11]. MS analysis can yield information on hundreds, or even thousands, of proteins, glycoproteins, and lipids, which can be correlated with known genomic findings to elucidate the underlying molecular mechanisms involved in tumorigenesis [12].

In a typical proteomics experiment, proteins to be analyzed are isolated from cell lysate or tissues by biochemical fractionation or affinity selection. In the next step, proteins are degraded enzymatically to peptides, usually by trypsin. Then, the peptides are separated by one or more steps of high-pressure liquid (nanoLC), eluted into an electrospray ion source, and the multiply protonated peptides enter the mass spectrometer where the *m*/*z* values for each peptide are detected (MS spectrum). When two or more peptides have the same *m*/*z* ratio but could have different primary sequences, for an improved identification, parent ions will be selected for fragmentation, and the MS/MS spectra will be recorded. In this way, even if the parent ions have an identical m/z, the generated fragments (daughter ions) are different for each parent ion due the structural differences. The MS and MS/MS spectra are matched against protein sequence databases. The outcome of the experiment is the identity of the peptides and therefore the protein sequence [13]. Electrospray ionization (ESI) and matrix-assisted laser desorption/ionization (MALDI) are the two techniques most commonly used to volatize and ionize the sample for the mass spectrometric analysis [14,15]. MALDI-MS is normally used to analyze relatively simple peptide mixtures, whereas integrated liquid-chromatography ESI-MS systems (LC-ESI-MS) are preferred for the analysis of complex samples [13]. The mass analyzer is characterized by some key parameters: sensitivity, resolution, mass accuracy and the ability to generate information-rich ion mass spectra from peptide fragments (tandem mass or MS/MS spectra). The mass analyzers currently used in proteomic research include linear ion traps (LIT), quadrupole ion traps (QIT) and mass filters (QMF), high-resolution Orbitraps, time-of-flight (TOF) and ion cyclotron resonance (ICR) mass spectrometers [16]. These mass analyzers are very different in design and performance, each with their own strength and weakness. These analyzers can be stand alone or, in some cases, put together in tandem to take advantage of each of their strengths [13].

The typical approach for a proteome study for biomarker discovery is to measure many proteins in various samples. The initial biomarker candidates are proteins with a different regulation pattern in patient samples compared to control samples [17]. Differentially regulated proteins identified by MS need to be further validated by an enzyme-linked immunosorbent assay (ELISA) or Western Blot analysis [18]. 

Proteomics studies in neuroscience progressed rapidly and allowed the development of a new field called “clinical neuroproteomics”, which aims to advance the understanding of diseases affecting the CNS through the study of the therapeutic and biomarker role of differentially expressed proteins in CNS tumors and to translate findings into clinical practice for routine screening, diagnosis and treatment development [19]. Therefore, several studies raised the possibility of developing a combined genomic and proteomic approach in order to personalize patient diagnosis and treatment [20].

The literature search for this review was conducted through the PubMed search engine using the keywords “diffuse glioma”, “astrocytoma”, “glioblastoma”, “ependymoma”, “oligoastrocytoma”, “proteomics”, and “mass spectrometry”. The initial results were further expanded with relevant literature found through the reference list of selected papers. The analysis only included English full-text papers published in the last ten years, although relevant papers have also been cited. The articles were first checked by title and abstract for relevance, then included based on the related findings to the topic. The aim of this review is to provide updates on the recent discoveries using classic and modern proteomic technologies, and to provide future research perspectives on glial tumors using MS approaches, focusing on clinically promising proteins involved in the pathogenesis of gliomas. 

## 2. Diffuse Astrocytic and Oligodendroglial Tumors

### 2.1. Diffuse Astrocytoma & Anaplastic Astrocytoma

WHO grade II and III diffuse gliomas of astrocytic morphology represent 14.1% of all gliomas [21]. According to the 2016 WHO classification of CNS tumors, astrocytomas are part of the diffuse astrocytic and oligodendroglial tumors containing two entities, the grade II diffuse astrocytoma & the grade III anaplastic astrocytoma. These entities are further subclassified, depending on the *IDH* gene status, into *IDH*-*mutated*, *IDH*-*wildtype* or *IDH not other specified (NOS)* when the genetic test could not be performed [1]. The articles included in this review refer to diffuse astrocytomas and anaplastic astrocytomas, independent of their *IDH* status, as many of the studies included in this review were performed before the wide adoption of the 2016 WHO classification.

The most frequent mutation in the *IDH1/2* genes is the *IDH1-R132H*, which is described in the majority of grade II and grade III gliomas [22,23]. A study conducted on 18 grade III anaplastic astrocytomas has shown that the presence of the *IDH1-R132H* mutation was associated with the up-regulation of a truncated C-terminal form of Alpha-crystallin B chain protein (CRYAB). The results were validated by an in vitro cell culture on the *IDH1*-*wildtype* and *IDH1-R132H* mutated cell lines, showing a significant increase of the CRYAB protein in the *IDH1-132H* mutated cell line [24]. CRYAB is a heat-shock protein involved in the proliferation, signaling, and anti-apoptotic activity of tumor cells [25]. Elevated levels of CRYAB can be detected in human cerebrospinal fluid (CSF) through MS, as it was reported for patients with multiple sclerosis [26]. Although this study suggests CRYAB as a possible indirect biomarker for *IDH1-R132H* mutation, further investigations are needed because the regulation difference could be caused by the grade difference between tumor samples, as *IDH-wildtype* anaplastic astrocytoma shares molecular features of the *IDH-wildtype* glioblastoma [1].

Radiotherapy plays a central role in the adjuvant treatment of astrocytomas. It is used to improve the overall survival and decrease recurrences [27]. In clinical practice, radiotherapeutic treatment can have divergent results on patients with apparently histologic identical astrocytomas, which raises the need for new biomarkers to guide clinicians toward the right treatment [28]. Phosphoglycerate kinase 1 (PGK1) and Cofilin-1 (COF1) proteins were up-regulated in radioresistant astrocytomas (Table 1) and were previously identified as biomarkers for an unfavorable prognosis in glioblastoma [25]. PGK1 is an enzyme involved in the metabolism of 1,3,di–phosphoglycerate to 3–phosphoglycerate, which up-regulates in gastric, prostate, endometrial, and pancreatic cancers and is correlated with a poor prognosis [29,30,31,32]. The up-regulation of PGK1 increases the glucose metabolism, which affects DNA replication and repair. In addition, PGK1 is known to support the progression and drug resistance of tumors [31,32]. 

COF1 is a protein involved in the dynamics of cytoskeleton actin filaments and the regulation of the cellular morphology and motility [53]. COF1 was found to be up-regulated in gastrointestinal endocrine tumors, colorectal cancer, and genitourinary tumors [54,55,56]. Therefore, PGK1 and COF1 should be further investigated as possible targets in tumor tissues for inducing radiosensitivity in glioma therapy [33].

Studies showed that Serum Albumin (ALBU) and Apolipoprotein A-I (APOA1) proteins were up-regulated in grade III astrocytomas, facilitating tumor proliferation through an increase in neovascularization and migration of tumor cells. Furthermore, these proteins accumulate in tumor cells, which could represent a cause of a putatively epileptogenic mechanism for long-term epilepsy-associated symptoms [34,57,58]. A further understanding of the mechanisms behind astrocytoma tumorigenesis are provided by MS, which identified the up-regulation in tumor tissue of Sorcin (SORCN) nitrated form and Tubulin beta chain (TBB5) nitrated form, which offers new insights into the tyrosine nitration events [35]. 

Pre-B-cell leukemia transcription factor-interacting protein 1 (PBIP1) is a protein involved in cell proliferation and tumor progression that was previously identified in pancreatic tumors and recently found to be up-regulated in astrocytomas and ependymomas, but not in oligodendrogliomas or the normal brain. This tumor specificity of PBIP1 may be used as a differential diagnostic biomarker between astrocytic and oligodendroglial tumors [36]. 

Fibulins are a family of seven proteins known to stabilize the structure of the extracellular matrix with important roles in the physiological functions of the cell, such as adhesion, migration, and proliferation [59]. The fibulin protein family gained attention for their role in tumorigenesis, being up-regulated in both tumor and stromal cells and influencing the composition of the tumor microenvironment [60,61]. Studies on astrocytoma cell lines identified three out of seven Fibulins (1, 2 and 5), differently regulated among grade I/II and grade III/IV astrocytoma, that could help us to understand the different evolution of astrocytic tumors among the four grades. Fibulin-1 (FBLN1) was described as being up-regulated in high-grade astrocytoma, and Fibulin-2 (FBLN2) and Fibulin-5 (FBLN5) was described as being down-regulated in grade II/III/IV compared with grade I astrocytoma [37].

### 2.2. Glioblastoma

Astrocytic tumors share some common proteins between tumor grades, but there are significant differences among them, with glioblastoma having the most specific proteomic signature [62]. Glioblastoma is the most aggressive tumor of the CNS, recently subclassified according to the *IDH* gene status into *IDH*-wildtype, *IDH*-*mutated* and *IDH*-*NOS* (1). More than 75% of glioblastomas are primary IDH-*wildtype* brain tumors, and only a few are secondary *IDH*-*mutated* glioblastoma. *IDH* mutation is strongly associated with secondary glioblastoma, as the majority of these glioblastoma are known to progress from lower grade diffuse gliomas (grade II/III Ast), thus having a better prognosis [63,64]. Other important individual prognostic factors are the age of diagnosis, the Karnofsky performance status and the methylation status of the *MGMT* promoter gene [65]. A comparative proteomic analysis of young and old glioblastoma patients identified multiple differentially expressed proteins involved in the regulation of the tumorigenesis that could explain the prognostic differences between young and old glioblastoma patients [38]. Among the differentially expressed proteins of special interest is the Phosphatidylethanolamine-binding protein 1 (PEBP1), an inhibitor to both Raf/MEK/ERK and nuclear factor kappa B pathways [66]. PEPB1 was up-regulated in young glioblastoma and down-regulated in old glioblastoma patients [38]. The regulation differences of PEPB1 between young and old glioblastoma patients should be further investigated to understand its influence on prognosis.

Neuronatin (NNAT) is a proteolipid membrane protein involved in the early development of the CNS that is endogenously down-regulated in the normal brain, and which becomes up-regulated in primary glioblastoma [39]. Recent findings indicate a NNAT up-regulation role in the progression of medulloblastoma and correlate it with a poor outcome in breast cancer [67,68]. NNAT implications in both health and disease processes highlight the need for more in-depth research toward an understanding of the roles and implications of the regulation differences of this protein [69].

A comparative proteomic analysis on 4 glioblastoma cell lines (U87, LN18, T98, and U118) was made to search for possible signatures for tumor invasiveness. A tumor cell invasion assay revealed the U87 cell line as the highest invasive cell line. The U87 cell line expression profile identified up-regulated proteins from the A Disintegrin and Metalloprotease (ADAM), Cathepsin, and Matrix metallopeptidase (MMP) families. These proteins were further analyzed using a template matching algorithm with the invasive phenotype of the four cell lines [70]. ADAM family proteins are a group of transmembrane and secreted proteins involved in fundamental processes of cellular homeostasis, including cell adhesion, migration, and signaling [71]. The dysregulation of the ADAM protein family, especially ADAM17, is involved in the development and progression of various cancers [72,73]. Cathepsins are a family of lysosomal proteases associated with various pathologic entities, including cancers that are being studied for their roles in the development and progression of malignancies [74,75]. MMP proteins are a family of endopeptidases involved in the remodeling of the extracellular matrix under physiological and pathological conditions, especially cancer, where these proteins were described to be up-regulated [76]. A further investigation of these protein families could help us understand the mechanisms behind the aggressive and invasive phenotype of glioblastoma.

Collagen alpha-1(VI) chain (CO6A1) was proposed as a tumor biomarker for glioblastoma, being up-regulated in glioblastoma and down-regulated in normal tissues [40]. CO6A1 accumulates and forms deposits in perivascular tumor tissue and in pseudopalisading cells, suggesting that it might be involved in the adaptation to hypoxia and have a role in tumor angiogenesis [40,77]. A proteomic analysis of glioblastoma angiogenesis confirmed that neoplastic blood vessels have a different proteomic signature than normal ones, with 29 proteins being specifically up-regulated in neoplastic vessels [78]. Moreover, altered extracellular matrix proteins in glioblastoma induce the up-regulation of Matrix metalloproteinase-9 (MMP9) and Metalloproteinase inhibitor 1 (TIMP1) proteins that disrupt normal angiogenesis and promote tumor invasion [37]. Annexin A2 (ANXA2) is a candidate biomarker for malignant gliomas that was additionally identified to be directly involved in angiogenesis-dependent invasion through the up-regulation of the vascular endothelial growth factor (VEGF) [79,80]. Neuron-glial-2 (NG2) is a transmembrane chondroitin sulphate proteoglycan that was previously correlated with a poor clinical outcome and up-regulated in glioblastoma, where it promotes drug resistance through PI3K/AKT survival signaling [41]. NG2 up-regulation in glioblastoma cells and blood vessels was proposed as an independent negative prognostic marker for patient survival [81]. An improved understanding of specific molecular characteristics involved in tumor angiogenesis is essential as it can offer new insights into tumor metabolism and reveal targets for anti-angiogenic interventions.

Exosomes are a promising source of molecular signatures for glioblastoma, being commonly isolated from CSF or from the fluid collected with the surgical aspirator during neurosurgical interventions [42]. Specific proteins isolated from glioblastoma exosomes are the invasion-related proteins (Annexin A1 (ANXA1), Insulin growth factor-2, Programmed cell death 6-interacting protein, Actin-related protein 3 and Integrin-β1), and the Polymerase 1 and transcript release factor complex (PTFR) [42,43]. PTFR is involved in the metabolic pathways of tumor cells and was associated with the chemoresistance of glioblastoma to imatinib [82,83]. PTFR is expressed in both glioblastoma tissue and serum exosomes; furthermore, in a murine model, silencing PTFR suppressed the glioma progression, suggesting that it could be a promising biomarker and a potential therapeutic target for glioblastoma [43]. Shen et al. identified 19 differentially regulated proteins in a systematic review conducted on CSF related glioma biomarkers, then, in an independent cohort, further validated 4 candidate proteins (Interleukin-6 (IL-6), Galanin peptides (GALA), Endoplasmic reticulum chaperone BiP (BIP), and Protein Wnt-4 (WNT4)) that could be studied to better understand how glioma cells interact and which pathways are activated in the local microenvironment of this tumor [44]. 

A new approach for reliable serum biomarkers is the possibility of using a combination of proteins instead of a single protein, as it is unlikely that one finds only a specific protein altered for a complex pathology such as glioblastoma [18]. An MS analysis on plasma collected from patients with glioblastoma identified several proteins that were differentially expressed in tumor tissue when compared to a normal brain, corresponding to the acute-phase reaction (Complement component C9, C-reactive protein, Alpha–1–antichymotrypsin, haptoglobin, ceruloplasmin, serum amyloid P, plasma retinol binding protein and α–1B–glycoprotein), lymphocytes cell signaling and immune response proteins, cell cycle regulation and proteins involved in the coagulation cascade [45,84,85,86]. Guanine nucleotide-binding protein G(o) subunit alpha (GNAO) is a promising diagnostic and prognostic marker for glioblastoma patients, GNAO increased plasma levels being correlated with a longer survival [45]. GNAO abnormal regulation levels influence the calcium flow in the brain tissue, the process being associated with the pathogenesis of epileptic encephalopathy [87]. A proteomic screening for potential protein patterns in glioblastoma patients serum revealed the alteration of several proteins (Bone morphogenetic protein 2 (BMP2), Platelet factor 4 (PLF4), C-X-C motif chemokine 10 (CXL10), Protein S100-A8 (S10A8), Protein S100-A9 (S10A9), Beta-Ala-His dipeptidase (CNDP1), Ferritin light chain (FRIL) and Heat shock 70 kDa protein family (HS71A, HS71B)) that have the potential to be used as biomarkers for a glioblastoma diagnosis or to be investigated as therapeutic targets [18,46,47]. 

Based on the proteomic alterations identified in glioblastoma, a network analysis can be used to identify protein models that include combinations of proteins, in order to better predict overall survival and to identify novel therapeutic targets [88,89,90]. 

### 2.3. Oligodendroglioma & Anaplastic Oligodendroglioma

Oligodendrogliomas are slow-growing, infiltrating brain tumors that have a better prognosis compared to other diffuse gliomas [1]. The oligodendroglial tumor entities present in the WHO 2016 CNS classification are the oligodendroglioma *IDH-mutated* 1p/19q co-deleted, oligodendroglioma NOS, anaplastic oligodendroglioma *IDH-mutated* 1p/19q co-deleted, anaplastic oligodendroglioma NOS and the highly debated entities oligoastrocytoma NOS and anaplastic oligoastrocytoma NOS, which, according to the new guideline, should be used only as exceptional exclusion diagnostics when genetic testing is not available [1]. The main genetic event present in OG is the 1p/19q co-deletion associated with a good response to chemotherapy, radiation therapy and an improved overall outcome [91].

Proteomics offer the possibility of understanding the phenotype induced by genetic events. A number of studies focused on decoding the proteomic differences of the 1p/19q co-deletion phenotype using proteomic technologies on previously diagnosed wildtype and co-deleted oligodendrogliomas. A pilot study on 5 matched 1p/19 co-deleted and 1p/19q non-deleted oligodendrogliomas revealed Brevican core protein (BCAN) and Serotransferrin (TRFE) as possible biomarkers for the co-deleted 1p/19q phenotype [48]. These proteins were identified by MS analysis and further validated by Western blot analysis. Another four proteins that were proposed to have the potential to be used as markers for the differentiation between oligodendrogliomas with and without 1p deletion are the V-type proton ATPase subunit E 1 (VATE1), Ubiquitin-like modifier-activating enzyme 1 (UBA1), High mobility group protein B1 (HMGB1) and Microtubule-associated protein 2 (MAP2) (Table 1). These proteins were studied on 47 oligodendroglioma samples and validated by both Western blot and immunohistochemistry. Further studies can unravel the different pathways that lead to a better prognosis and chemosensitivity in the OG 1p loss of the heterozygosity (LOH) phenotype [49]. 

The proteomic analysis of the differential regulation of proteins in a case of rapid progression from grade II oligodendroglioma to anaplastic oligodendroglioma displayed an abnormal regulation level of Peroxiredoxin 6 (PRDX6) and Rho GDP-dissociation inhibitor alpha (GDIR1), which can be further investigated as candidates for molecular predictive factors of malignant transformation [50]. 

## 3. Other Astrocytic Tumors

Pilocytic astrocytoma, subependymal giant cell astrocytoma, pleomorphic xanthoastrocytoma and anaplastic pleomorphic xanthoastrocytoma are primary CNS tumors that mainly occur in children and young adults and are usually associated with a favorable outcome [92]. Even if the prognosis of these tumors is better when compared to adult astrocytomas, pediatric brain tumors remain the leading cause of childhood cancer deaths [92,93]. 

A preliminary study used an integrated proteomic approach for the study of pilocytic astrocytoma and identified regulation differences of several proteins, such as actin, Ig kappa chain C region, Serotransferrin, Tubulin beta 2A chain and Vimentin (VIME) (Table 1) [94]. VIME is a member of the intermediate filament family of proteins, with roles in the maintenance of cellular integrity, and it was found to be up-regulated in gastric, prostate, breast and lung cancer. VIME up-regulation was frequently associated with tumor growth and poor prognosis [95]. The VIME role in the pathogenesis of pediatric astrocytomas is still debated, and more studies need to validate this protein as a possible biomarker [51]. Different regulation levels among pediatric astrocytoma grades were described for several proteins, including Calreticulin (CALR) and 14-3-3 protein epsilon (1433E) and for 22 miRNA, among which 12 were involved in the glycosaminoglycan biosynthesis, a pathway that was previously described in the invasion and progression of glioblastoma [51]. Although some promising proteins were recently characterized, more studies with bigger cohorts of pediatric CNS neoplasms are needed for the postulation of prognostic or diagnostic biomarkers.

## 4. Ependymal Tumors

Ependymomas are a group of primary intracranial neuroepithelial tumors that occur both in pediatric and adult patients, and they are frequently located in the posterior fossa and spinal cord [52,96]. Among ependymal tumors, pediatric ependymomas are the third most common pediatric tumors. Currently, the diagnosis and prognosis of ependymomas are solely based on histological and clinical criteria [93,96]. Thus, research efforts have focused on studying the underlying proteomic landscape of these tumors. The Pediatric Ependymoma Protein Database (PEPD) was built by MS analyses of WHO grade II ependymomas. PEPD includes data on more than 5000 proteins and 15,675 peptides organized in 4157 protein groups [93]. This database is a valuable research tool as it provides comprehensive information on the proteomic landscape of ependymal tumors, which can be analyzed for a better understanding of the molecular processes that take place in these tumors. 

Novel insights into ependymal tumor proteomics are represented by ANXA1 and Calcyphosin (CAYP1), two calcium binding protein that were significantly up-regulated in ependymomas, the regulation level being correlated with the mRNA expression level and confirmed by immunohistochemistry [52]. Additionally, ANXA1 was described as being up-regulated in the tumor neovascular endothelium, and targeting ANXA1 with antibodies resulted in the improvement of the overall survival and response to radiotherapy [97]. ANXA1’s role in cancer was previously described in many oncogenic processes, from cancer progression, metastasis, and resistance to therapy [98,99]. The CAYP1 regulation level in ependymoma does not correlate with the tumor grade or localization but was up-regulated in ependymomas with epithelial differentiation, suggesting that it is a potential biomarker for a new sub-category of ependymomas [52]. CAYP1 up-regulation was associated with a poor prognosis in colon cancer, squamous cell carcinoma and breast cancer [100,101,102]. In breast cancer, calcyphosin was described as a predictive marker for tamoxifen resistance; as a result, its role in promoting resistance to therapy should be further investigated in other tumors [102]. 

## 5. Conclusions

Proteomic technologies have the potential to improve our understanding of the intrinsic mechanisms and to uncover pathways and means of communication between tumor cells, the microenvironment, and the organism by adding a new perspective on the molecular landscape of brain tumors. The MS-based proteomic approaches enabled the characterization of the proteome of glial tumors. The new high-end hybrid tandem mass spectrometers allowed for the integration of large amounts of data, which is required to handle the intratumoral heterogeneity present in glial tumors. Intratumoral heterogeneity is an important aspect that needs consideration for the decoding of the specific proteomic and genomic landscape of individual cell populations from the same tumor to personalize the diagnosis of tumor entities. Glial tumors and glioblastoma in particular are known to consist of different cell populations (tumor stem cells or more differentiated tumor cells), with different activated signaling pathways, having different biological behaviors and different responses to treatment. Moreover, considering the sensitivity of proteomic analyses, the preanalytical phase parameters (cold and warm ischemia, storage conditions and sample processing) have to be taken into account in order to obtain valuable and reproductible results. However, there is still a need for standardization in the MS analysis of brain tumors in terms of the control sample used, MS protocols, sample processing techniques, and bioinformatics analysis [58]. Data should be interpreted with caution, given that this information could be useful only if other molecular parameters, such as the genetic profile, microRNA or long-noncoding RNA, are factored in. Today, the *IDH* gene status or the methylation of the promoter of the *MGMT* gene have a prognostic and predictive role, and should be considered possible confounders in statistical analyses when trying to determine the importance of a protein marker in the biology of a glial tumor. Alongside clinical data, such as age, tumor size and location or Karnofsky Score, protein parameters will shed light on the new possible druggable targets.

Currently, there is a need for the clinical validation of biomarkers with bigger cohorts and by independent research teams in order to avoid reproducibility gaps. For example, Chitinase–3–like protein 1 (CHI3L1) was the highest expressed protein in the U87 cell line and in the same study was further validated by Western Blot and correlated with cell line invasiveness [70]. However, despite initial promising results, the CHI3L1 role could not be further validated [103,104]. For biomarker discovery, targeted approaches are expected to bridge the gap between candidate discovery and the clinical assays development of validated biomarkers [105].

The majority of MS studies generate huge amounts of proteomic data that identify numerous differentially regulated proteins among tumor entities, but they lack the integration of the previous genomic information. Currently, there is a need for a holistic approach that integrates the data generated by different omics approaches (genomics, proteomics, metabolomics) in order to provide a complete view of the complex processes that take place at the cellular level in glial tumors. These approaches will ultimately change the current diagnostic standards of brain tumors from optic microscopy to an integrated approach that combines histologic, genomic and proteomic data. 

## Figures and Tables

**Table 1 medicina-55-00412-t001:** Regulation status of proteins identified by proteomic approaches.

Tumor	Protein	Methods	Samples	Control	Regulation Status ↑/↓	Ref.
Ast Gr III	CRYAB	MALDI-TOF/TOF	9 Ast Gr III–IDH1-R132H mutant	9 Ast Gr III–IDH1-R132H wildtype	↑ IDH1-R132H mutant Ast	[24]
Ast Gr II	PGK1	2D-LC–MS/MS, WB	8 Ast Gr II, radioresistant	7 Ast Gr II, radiosensitive	↑ radioresistant Ast Gr II	[33]
Ast Gr II	COF1	2D-LC–MS/MS, WB	8 Ast Gr II, radioresistant	7 Ast Gr II, radiosensitive	↑ radioresistant Ast Gr II	[33]
Ast Gr I–IV	ALBU	2DGE	tumor tissue	normal brain tissue	↑	[34]
Ast Gr I–IV	APOA1	2DGE	tumor tissue	normal brain tissue	↑	[34]
Ast Gr. I–IV	SORCN	2DGE, LC-MS/MS, 2D WB	18 Ast Gr I–IV tissue	normal brain tissue	↑	[35]
Ast Gr. I–IV	TBB5	2D PAGE, LC-MS/MS, 2D WB	18 Ast Gr I–IV tissue	normal brain tissue	↑	[35]
Ast Gr II, III	PBIP1	in silico proteomics IHC IFM qRT-PCR	95 Ast Gr II, III tissue	normal brain tissue	↑	[36]
Ast Gr. I–IV	FBLN1	2D-LC–MS/MS iTraQ	5 GB tissue	15 Ast Gr I–III tissue	↑	[37]
Ast Gr. I–IV	FBLN2	2D-LC–MS/MS iTraQ	5 Ast Gr I	15 Ast Gr II–IV	↑	[37]
Ast Gr. I–IV	FBLN5	2D-LC–MS/MS iTraQ	5 Ast Gr I tissue	15 Ast Gr II–IV tissue	↑	[37]
GB	MMP9	2D-LC–MS/MS iTraQ	5 GB tissue	15 Ast Gr I–III tissue	↑	[37]
GB	TIMP1	2D-LC–MS/MS iTraQ	5 GB tissue	15 Ast Gr I–III tissue	↑	[37]
GB	PEBP1	LC–MS/MS	7 young GB tissue	12 young peritumoral tissue	↑	[38]
			13 old GB tissue	10 old peritumoral tissue	↓	
			13 old GB tissue	7 young GB tissue	↓	
GB	NNAT	nCIF LC–MS/MS	59 GB tissue	normal brain tissue	↑	[39]
GB	COL6A1	LC–MS/MS	38 GB tissue	normal brain tissue	↑	[40]
GB	NG2	2DGE, LC–MS/MS	96 GB tissue	96 GB tissue	↑ poor prognosis GB	[41]
GB	ANXA1	2D-LC–MS/MS	U87 cell line exosomes	LN229 cell line exosomes	↑	[42]
GB	PTFR	WB IHC	4 primary GB cell lines U87, LN229, U251 cell lines, 36 GB tissue		↑ poor prognosis GB	[43]
GB	IL-6	ELISA	9 GB CSF & CF	6 non-tumor CSF 11 Gr I–III glioma CSF &CF	↑ GB CF &CSF	[44]
GB	GALA	ELISA	9 GB CSF & CF	6 non-tumor CSF 11 Gr I–III glioma CSF &CF	↓ GB CF	[44]
GB	BIP	ELISA	9 GB CSF & CF	6 non-tumor CSF 11 Gr I–III glioma CSF &CF	↓GB CF	[44]
GB	WNT4	ELISA	9 GB CSF & CF	6 non-tumor CSF 11 Gr I–III glioma CSF &CF	↓ GB CF	[44]
GB	GNAO	WB	17 GB serum	17 controls serum	↑ GB serum	[45]
GB	CXL10	ELISA/IHC	23 GB serum	12 controls serum	↓ GB serum	[46]
GB	BMP2	ELISA/IHC	23 GB serum	12 controls serum	↑	[46]
GB	HSP71A, HS71B	ELISA/IHC	23 GB serum	12 controls serum	↑	[46]
GB	PLF4	SELDI-TOF MS, LC-MS/MS	35 GB serum	30 controls serum	↑	[18]
GB	S10A8	SELDI-TOF MS, LC–MS/MS	35 GB serum	30 controls serum	↑	[18]
GB	S10A9	SELDI-TOF MS, LC–MS/MS	35 GB serum	30 controls serum	↑	[18]
GB	FRIL	LC–MS/MS ELISA	23 GB serum	12 controls serum	↑	[47]
GB	CNDP1	LC–MS/MS ELISA	23 GB serum	12 controls serum	↓	[47]
OG	BCAN	LC–MS WB	5 OG 1p/19q undeleted tissue	5 OG 1p/19q co-deleted tissue	↑	[48]
OG	TRFE	LC–MS WB	5 OG 1p/19q undeleted tissue	5 OG 1p/19q co-deleted tissue	↑	[48]
OG	VATE1	2D-LC/MS/MS iTraQ	4 OG 1p LOH tissue	4 OG 1p non-deleted tissue	↑	[49]
OG	UBA1	2D-LC/MS/MS iTraQ	4 OG 1p LOH tissue	4 OG 1p non-deleted tissue	↓	[49]
OG	HMGB1	2D-LC/MS/MS iTraQ	4 OG 1p LOH tissue	4 OG 1p non-deleted tissue	↑	[49]
OG	MAP2	2D-LC/MS/MS iTraQ	4 OG 1p LOH tissue	4 OG 1p non-deleted tissue	↑	[49]
OG	PRDX6	MALDI-TOF	OG tissue	AO recurrence tissue	↑ OG ↓ AO	[50]
OG	GDIR1	MALDI-TOF	OG tissue	AO recurrence tissue	↓ OG ↑ AO	[50]
PA	VIME	2DGE MALDI-TOF	3 PA	4 normal brain tissue	↑	[51]
PA	CALR	2DGE MALDI-TOF	3 PA	4 normal brain tissue	↑	[51]
PA	1433E	2DGE MALDI-TOF	3 PA	4 normal brain tissue	↑	[51]
Ependymoma	PBIP1	WB qRT-PCR	12 Ependymoma Gr II/III tissue	normal brain tissue	↑	[36]
Ependymoma	ANXA1	MALDI-TOF	12 ependymomas Gr II/III tissue	7 normal brain tissue	↑	[52]
Ependymoma	CAYP1	MALDI-TOF MS/MS	12 Ependymomas Gr II/III tissue	7 normal brain tissue	↑	[52]

Ast Gr II: Astrocytoma Grade II; Ast Gr III: Astrocytoma Grade III; AO: Anaplastic Oligodendroglioma; CF: Cystic Fluid; CSF: Cerebrospinal Fluid; GB: Glioblastoma; LOH: Loss of heterozygosity; OG: Oligodendroglioma; PA: Pilocytic astrocytoma; LC–MS/MS: liquid chromatography-tandem mass spectrometry; 2LC-ESI–MS/MS: two dimensional liquid chromatography-tandem mass spectrometry; MALDI-TOF: matrix-assisted laser desorption/ionization (MALDI) and time-of-flight (TOF) mass analyzer; SELDI-TOF: surface-enhanced laser desorption/ionization (SELDI) and time-of-flight (TOF) mass analyzer; 2DGE: two dimensional gel electrophoresis; ELISA: enzyme linked immunosorbent assays; WB: Western Blot analysis; IHC: Immunohistochemistry; iTRAQ: Isobaric tags for relative and absolute quantitation; IFM: Immunofluorescence microscopy; and nCIF: nano-capillary isoelectric focusing.

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
