# Peer review of "Proteomic Advances in Glial Tumors through Mass Spectrometry Approaches"

_1010-660X, 2019, doi:10.3390/medicina55080412_

Round 1

Reviewer 1 Report

This is an original,  concise but thorough overview of findings amassed from studies of the proteome of tissue and plasma from gliomas  (GBM, oligodendrogliomas, astrocytomas and ependymomas). The work is well structured and easy to read. The strength of the work is the inclusion of studies that  well characterised their samples in terms of diagnostic molecular pathology. IT becomes even clearer that although we know a lot about biological  behaviour  and perhaps eitiology  of these tumours, we still do not have enough actionable options  for treatment. 

     Where possible the authors have taken the extra step to explain the functional and clinical relevance of the salient proteins that were found to be  differentially expressed in the tumours and  attempted to clarify the differences based on distinguishing molecular pathology of the tumours. However, with the exception of 4 lines in the discussion section (318-321), the discussion is largely uncritical and at times comes across as summative. For example, it would have been informative to see a specific critical discussion of studies that link prognostication to particular proteins, whether these were independently prognostic, controlled  for known factors such as age, MGMT status, KPS. It is understandable that molecular pathology was not systematically available in older studies. However, it  would have been good to address statistical robustness regarding  sample sizes, location of tumour, comment regrading cellular heterogeneity and inter/intra sample variability, particularly for GBM tumours, and how interpretation of results  might be  accounted for  in this  regard in the various studies.

A good example for this is illustrated in table 1 for the correlation of CRYAB protein and IDH-R132H mutation .  The conclusion of this particular study (ref 24) could simply be that the proteomic profile was detecting differences due to tumour grade, rather than aggressiveness as such. Since the samples were effectively grade III Astr (IDHmut) vs GBM (IDHwt). I think a slightly more critical discussion of the methodology and statistical validation would make the work even more interesting. Case in point,  discussion of findings ( line181-184); U87 is a homogenous  cell line, originally derived from a grade III astrocytoma. After many years in 2D- subculture, this cell line is homogenous and  does not by any means display invasive growth.

Reviewer 2 Report

In this paper by Pirlog et al the authors review the proteomics data obtained by mass spectroscopy technology in relation to glial tumors. The paper is of interest and nicely summarizes the most recent and pertinent findings. The manuscript needs some proofreading for English language grammar and style. Major and minor comments are listed below:

Major:

1.      Authors should clarify in abstract and introduction what types of gliomas they will discuss (in this case diffuse gliomas, pilocytic astrocytomas, ependymomas).

2.      Authors should offer in introduction the timeframe for the literature search, if search was performed only in the English literature or Other, and what were the key words used for search.

3.      Line 36 - 37 – IDH and TERT promoter mutations are not encountered in WHO grade I gliomas. Please correct.

4.      Lines 61-62 – "For an improved identification a given peptide ion could be selected for fragmentation MS/MS spectrum are recorded." – clarification needed.

5.      Line 92-93 – do authors mean 50% of all primary brain tumors?

6.      Lines 103-104 – authors should expand on the sensitivity and specificity (accuracy) of detecting CRYAB levels in CSF and the presence of IDH1 mutation. Also only IDH1-R132H mutations or other IDH mutations as well? How many samples were used etc.

7.      Table 1 – number of samples used should be included for every citation included in table

8.      Line 164 – "lower grade gliomas" should be "lower grade diffuse gliomas"

9.      Line 247 – "other gliomas" should be "other diffuse gliomas"

10.   Lines 252-253 – 1p/19q codeletion is present in 100% of oligodendrogliomas. If a tumor does not have the co-deletion it is not an oligodendroglioma. Please rephrase. Also please refer to this as "co-deletion" not as "mutation".

11.   Expand on the sensitivity and specificity of protein biomarkers for identification of 1p/19q co-deletion. Include number of samples studied so far etc. (lines 255-262)

12.   Line 278 – please specify the cancers you are referring to and note that epithelial to mesenchymal transition is not a feature of the CNS primary brain tumors.

13.   Why authors only refer to pediatric ependymomas (line 287)? A justification should be provided.

14.   Suggest eliminating the Discussion section and add the paragraph under Conclusions.

Minor:

1.      Line 37 - "TERT" should be "TERT promoter"

2.      Line 53 – "multiple entities" should be "multiple proteins"

3.      Line 60 – define/explain "m/z values"

4.      "ESI" abbreviation appears twice – please double check

5.      Line 79 – Western blot and immunoblot are the same thing. Please check.

6.      Check if all proteins are defined – line 181 – ADAM, MMP not defined. Suggest including a list with abbreviations at the end of the manuscript.

7.      Line 271 – "adult Ast"? Was defined in table 1 but not in main text.

Round 2

Reviewer 2 Report

The authors nicely addressed all comments and the paper has been significantly improved. Following second review, a few issues/clarifications remain. Few very minor issues were also identified.  

1. Introduction - "Previous research focused on decoding the genetic alterations behind these tumors with important mutations in PTEN, EGFR and BRAF genes being observed in grade III and grade IV gliomas and mutations in IDH-1, IDH-2, ATRX, 1p-19q, TERT promoter genes being frequently identified in grade II gliomas" – Needs to be rephrased as it offers incomplete and misleading information. IDH1/2, ATRX, 1p/19q co-deletion, TERT promoter are not only present in grade II gliomas. They are also present in grade III and IV gliomas. Also BRAF is common in PXA, ganglioglioma, pilocytic, glioblastoma but not in anaplastic astrocytoma. Please rephrase entire sentence.

2. Please use italics when referring to genes (i.e. IDH1, IDH2, EGFR, BRAF) and keep regular text format (not italics) when referring to a protein or family of genes (i.e. IDH, IDH-mutant (gene family), BRAF (protein), COF1, PTEN, etc). This is the standard nomenclature for genes and proteins in humans (https://academic.oup.com/molehr/pages/Gene_And_Protein_Nomenclature).

3. " Diffuse astrocytomas, including both diffsue and anaplastic astrocytoma, represents 7.4% of the

primary brain tumors, having the second highest incidence after glioblastoma". This statement is incorrect. Per Reference #21 the authors cite, Figure 7A, Diffuse astrocytomas (aka WHO grade II) represent indeed 7.4% of all gliomas (not of all primary brain tumors). Also this category does not include "both diffuse and anaplastic astrocytomas". Would be more correct to state that WHO grade II and III diffuse gliomas of astrocytic morphology represent 14.1% (7.4+6.7%) of all gliomas. Please correct statement.

4. Line 116 – " IDH1-wt" should be "IDH1-R132H-wt"? (were cell lines tested for all IDH1 mutations, aka was the entire gene sequenced, or just for R132H? – please check original reference and update accordingly)

Minor:

1. Typo – " diffsue" (line 100)

2. Line 30 – add ", and ependymomas."

3. Line 28 - add comma between "findings" and "possible"

4. Line 48 "fill the gap'

5. Line 94 – ", and ,"mass spectrometry"".

6. Line 108 "not otherwise specified (NOS)" is correct.

7. Line 162 – "used as a differential diagnostic" is correct
